# Effects of Electrochemotherapy on Immunologically Important Modifications in Tumor Cells

**DOI:** 10.3390/vaccines11050925

**Published:** 2023-04-30

**Authors:** Ursa Kesar, Bostjan Markelc, Tanja Jesenko, Katja Ursic Valentinuzzi, Maja Cemazar, Primoz Strojan, Gregor Sersa

**Affiliations:** 1Department of Experimental Oncology, Institute of Oncology Ljubljana, 1000 Ljubljana, Slovenia; 2Faculty of Medicine, University of Ljubljana, 1000 Ljubljana, Slovenia; 3Biotechnical Faculty, University of Ljubljana, 1000 Ljubljana, Slovenia; 4Faculty of Health Sciences, University of Primorska, 6310 Izola, Slovenia; 5Department of Radiation Oncology, Institute of Oncology, 1000 Ljubljana, Slovenia; 6Faculty of Health Sciences, University of Ljubljana, 1000 Ljubljana, Slovenia

**Keywords:** electrochemotherapy, cisplatin, oxaliplatin, bleomycin, immune response, immunogenic cell death, ATP, HMGB1, calreticulin

## Abstract

Electrochemotherapy (ECT) is a clinically acknowledged method that combines the use of anticancer drugs and electrical pulses. Electrochemotherapy with bleomycin (BLM) can induce immunogenic cell death (ICD) in certain settings. However, whether this is ubiquitous over different cancer types and for other clinically relevant chemotherapeutics used with electrochemotherapy is unknown. Here, we evaluated in vitro in the B16-F10, 4T1 and CT26 murine tumor cell lines, the electrochemotherapy triggered changes in the ICD-associated damage-associated molecular patterns (DAMPs): Calreticulin (CRT), ATP, High Mobility Group Box 1 (HMGB1), and four immunologically important cellular markers: MHCI, MHC II, PD-L1 and CD40. The changes in these markers were investigated in time up to 48 h after ECT. We showed that electrochemotherapy with all three tested chemotherapeutics induced ICD-associated DAMPs, but the induced DAMP signature was cell line and chemotherapeutic concentration specific. Similarly, electrochemotherapy with CDDP, OXA or BLM modified the expression of MHC I, MHC II, PD-L1 and CD40. The potential of electrochemotherapy to change their expression was also cell line and chemotherapeutic concentration specific. Our results thus put the electrochemotherapy with clinically relevant chemotherapeutics CDDP, OXA and BLM on the map of ICD inducing therapies.

## 1. Introduction

Nowadays, different ablative techniques are used in clinic for tumor treatment [1]. Electrochemotherapy (ECT) is a local ablative technique, mainly employed in treatment of superficial tumors, with published clinical guidelines supporting its use in melanoma, squamous cell carcinoma, including its management in patients with epidermolysis bullosa, breast cancer, Merkel cell carcinoma, soft tissue sarcomas and bone metastases [2,3]. The technique can also be used for the treatment of deep-seated tumors [4,5]. In electrochemotherapy, electrical pulses are used to increase cell membrane permeability and allow targeted entry of chemotherapeutic drugs, such as Cisplatin (CDDP), Oxaliplatin (OXA) or Bleomycin (BLM) into the cell, potentiating their cytotoxicity at the application site [2,6]. The main reason for cell death after electrochemotherapy with CDDP, OXA and BLM is DNA damage due to formation of intrastrand dinucleotide DNA adducts (CDDP, OXA) or induced double-strand breaks (BLM) [7,8,9]. Besides these direct cytotoxic effects, electrochemotherapy can also trigger indirect antitumor effects with the activation of the immune system through the antigen shedding from dying tumor cells and immunogenic cell death (ICD) [10,11].

ICD is a cell death modality that effectively stimulates an adaptive immune response against neo-antigens released by dying or dead cells [12,13]. Thus, dying cancer cells are converted into an anticancer vaccine [10]. The immunogenic characteristics of ICD are mainly mediated by the release of molecular signals in response to cell death and stress, and are called damage-associated molecular patterns (DAMPs) [14,15]. The most relevant ICD-associated DAMPs considering the clinical response of tumor cells are (1) the secretion of adenosine triphosphate (ATP); (2) the extracellular release of High Mobility Group Box 1 (HMGB1); and (3) the exposure of endoplasmic reticulum chaperone Calreticulin (CRT) on cell membrane [16,17].

During the course of ICD, intracellular metabolite ATP is released in an autophagy-dependent manner through the active exocytosis [18]. ATP release is mediated by active secretion from dying tumor cells preceding the release of HMGB1 [14,19]. Extracellular ATP is a chemoattractant for dendritic cells (DCs), their precursors and macrophages [10,18]. It facilitates the recruitment of myeloid cells and enables their differentiation into mature DCs when binding to P2Y2 purinergic receptors [18]. The immunogenicity of cell death is abrogated when either ATP fails to accumulate in the microenvironment of dying tumor cells or when purinergic receptors are absent from myeloid cells [18].

ICD is also associated with the release of non-histone chromatin-binding protein HMGB1, that acts as a DNA chaperone [19,20]. It can be actively secreted via mechanism that require posttranslational modifications from the nucleus to the cytosol, or it can be passively released by necrotic or damaged cells into the extracellular space [20,21]. There it binds to Toll-like receptor 4 (TLR4) on DCs, stimulating their antigen-presenting functions and activating the release of proinflammatory cytokines from monocytes/macrophages [22,23]. HMGB1 triggers proinflammatory responses by signaling through receptors on the surface of both immune and nonimmune cells [23].

CRT is an endoplasmic reticulum (ER)-associated chaperone that is the main regulator of Ca^2+^ homeostasis and is also responsible for loading of cellular antigens into major histocompatibility complex (MHC) class I molecules [15,24]. In stressed or dying cells CRT translocates from the ER to the cell surface: its exposure serves as a co-stimulatory and pre-mortem (“eat-me”) signal to antigen presenting cells (APC) and immune cells, such as natural killer (NK) cells [15,19,24]. Thus, it is essential for the elicitation of an immune response [10].

Besides ICD-associated DAMPs, other immunologically important changes in tumor cells contribute to the immune response as well. Tumor cells can avoid tumor-associated antigen presentation by downregulation of surface display of MHC I, which is a crucial factor in the initiation of adaptive immune response [25]. MHC I is expressed on all nucleated cells and presents endogenously-derived peptide antigens—A process important for reporting intracellular changes—to the immune system through T-cell receptors (TCRs) of CD8+ T-cells [25]. On the other hand, MHC II molecules are normally expressed in professional APCs and present exogenously-derived peptide antigens to CD4+ T-cells of the immune system [26]. However, many other cell types express MHC II, including some tumor cells: thus, tumor-specific MHC II expression may increase tumor recognition by the immune system [27,28]. Expression of MHC II in general correlates with improved treatment outcome [28,29]. 

Tumor cells can also evade immune surveillance by binding of programmed cell death ligand-1 (PD-L1), which is expressed on tumor and other cells in the tumor microenvironment, to programmed cell death protein-1 (PD-1) expressed on T cells [29]. This results in the inhibition of TCR signaling, while prolonged PD-1/PD-L1 engagement creates hypofunctional, “exhausted” T cell state, that enables tumor progression [29].

Another immunologically important molecule is CD40, which is a member of the TNF receptor family, and a co-stimulatory cell surface receptor, expressed by APCs, other non-immune cells (e.g., epithelial and endothelial cells, platelets, fibroblasts, etc.) and tumor cells [26,30,31]. When CD40 binds to its ligand CD40L, this interaction mediates anti-tumoral immune responses by increasing ICD of tumor cells, upregulating the expression of MHC molecules and producing proinflammatory factors [30,31].

It was previously shown that electrochemotherapy with BLM can induce ICD; however, whether electrochemotherapy with other chemotherapeutic drugs, such as CDDP or OXA, also induces ICD and if the phenomenon is ubiquitous across different concentrations and different tumor cell lines is unknown [32]. In regards to the induction of DAMPs, it was shown that BLM can induce the release of ATP, the translocation of CRT, and the release of HMGB1; similarly, CDDP can induce the release of ATP and HMGB1, and OXA can induce the release of ATP, the translocation of CRT, and the release of HMGB1 and HSP70 [10,33,34,35].

Besides DAMPs, chemotherapeutics can also help upregulate antigen presentation and decrease tumor immune escape [25,36]. It was already shown that in some cancers CDDP can upregulate MHC I and PD-L1; OXA can upregulate MHC I and PD-L1; and BLM can upregulate PD-L1 [34,36,37,38].

Therefore, in this study we focused on the evaluation of the induction of ICD-associated DAMPs: ATP, HMGB1 and CRT and cellular markers MHC I, MHC II. PD-L1, and CD40 that can contribute to immunologically important changes in tumor cells in vitro after electrochemotherapy with BLM, CDDP or OXA. The potential of electrochemotherapy with BLM, CDDP or OXA at three different concentrations, i.e., IC_30_, IC_50_ and IC_70_, and at different time points after electrochemotherapy was determined in three murine cell lines forming immunologically distinct tumor models in vivo, i.e., B16-F10 melanoma (poorly immunogenic), 4T1 breast cancer (immunogenic) and CT26 colon cancer (highly immunogenic tumor) models. 

## 2. Materials and Methods

### 2.1. Cell Lines and Drugs

All cell lines were purchased from American Type Culture Collection (ATCC, Manassas, VA, USA). The B16-F10 murine melanoma cells were cultured in Advanced DMEM (Gibco, Thermo Fisher Scientific, Waltham, MA, USA), whereas 4T1 murine mammary carcinoma cells and CT26 murine colorectal carcinoma cells were cultured in Advanced RPMI-1640 (Gibco). Media were supplemented with 5% (*v*/*v*) fetal bovine serum (FBS, Gibco), GlutaMAX (100×, Gibco), and penicillin-streptomycin (100×, Sigma-Aldrich, Merck, Darmstadt, Germany). All cells were maintained in a humidified incubator with 5% CO_2_ at 37 °C. The cells were routinely tested for mycoplasma infection by MycoAlert^TM^ PLUS Mycoplasma Detection Kit (Lonza, Basel, Switzerland) and were mycoplasma free.

CDDP (1 mg/mL, Accord Healthcare Ltd., London, UK) was diluted in Advanced RPMI-1640 with GlutaMAX, with penicillin-streptomycin and without FBS to working solutions: 400, 300, 200, 150, 100, 50 and 25 µM. OXA (5 mg/mL, Teva Pharmaceutical Industries Ltd., Jerusalem, Israel) was diluted in Advanced RPMI-1640 with GlutaMAX, with penicillin-streptomycin and without FBS to working solutions: 800, 600, 400, 300, 200, 150, 50 and 25 µM. BLM (Bleomycin sulfate, 10 mg, Selleckchem, Houston, TX, USA) was diluted in Advanced RPMI-1640 with GlutaMAX, with penicillin-streptomycin and without FBS to working solutions: 18, 13, 10, 5, 1 and 0.1 nM.

### 2.2. Determination of Electropermeabilization

Cells were trypsinized, centrifuged and resuspended in cell medium without FBS. For each experimental group 2 × 10^6^ cells were resuspended in 90 µL of medium without FBS and 10 µL of 100 µM Propidium Iodide (PI) solution (Sigma-Aldrich). 50 µL of the mixture was pipetted between two stainless steel plate electrodes (2.4 mm distance between the electrodes) and electroporated with 8 square wave pulses, with 100 µs duration at a frequency of 1 Hz. Electric pulse generator (Jouan GHT beta, LEROY Biotech, Saint-Orens-de-Gameville, France) was used to deliver different amplitude per distance ratios to each group: 0, 300, 500, 700, 900, 1100, 1300 and 1500 V/cm. Immediately after electroporation, the cell mixture was transferred into the wells of a 24-well ultra-low attachment plate (Corning Costar, Corning, NY, USA), and 5 min later, 2 mL of cell culture medium was added. The cells were then transferred into 5 mL round-bottom polystyrene test tube (Falcon, Corning), centrifuged and washed twice in 1× phosphate-buffered saline (PBS) with 5% FBS, and analyzed the percent of PI positive cells with FACSCanto II Flow cytometer (BD Biosciences, San Jose, CA, USA). The data were then analyzed using FlowJo software (Tree Star Inc., Ashland, OR, USA).

### 2.3. Electrochemotherapy (ECT)

Cells were trypsinized, centrifuged and resuspended in cell medium without FBS. Each experimental group contained 2 × 10^6^ cells in 90 µL of cell medium without FBS and 10 µL of CDDP, OXA or BLM solution or 10 µL of cell medium without FBS in the case of untreated or electroporation only controls. The final concentrations of cytotoxic drugs were therefore 10 times lower than the working concentrations (CDDP: 40, 30, 20, 15, 10, 5, 2.5 µM; OXA: 80, 60, 40, 30, 20, 15, 5, 2.5 µM; BLM: 1.8, 1.3, 1, 0.5, 0.1, 0.01 nM). Then, 50 µL of the mixture containing 1 × 10^6^ cells was pipetted between two stainless steel plate electrodes (2.4 mm distance between the electrodes) and electroporated with 8 square wave pulses, 1300 V/cm, 100 µs duration at frequency of 1 Hz with electric pulse generator (Jouan GHT beta). The cell mixture was then transferred into the wells of a 24-well ultra-low attachment plate (Corning Costar), and 5 min after electric pulse delivery, 2 mL of cell culture medium was added. The other 50 µL of the mixture was not electroporated and only transferred into the wells of a 24-well ultra-low attachment plate (Corning Costar), and after 5 min 2 mL of cell culture medium was added.

### 2.4. Clonogenic Assay

After electrochemotherapy, cells were seeded at different densities (Appendix A) in 6-well plates (VWR International) in triplicate and incubated for six days at 37 °C in a 5% CO_2_ humidified incubator until colony counting. After six days the colonies were stained with crystal violet solution in 99.8% methanol (Sigma-Aldrich) and colonies containing at least 50 cells were counted. Plating efficiency (PE—Number of counted colonies divided by the number of seeded cells) and Surviving fraction (SF—Treated group PE divided by the average PE of the Control group) were calculated and the SF curve was plotted in GraphPad Prism (GraphPad Prism 9 Software, San Diego, CA, USA). From the SF curve, the inhibitory concentrations for each drug reducing cell survival by 30% (IC_30_), 50% (IC_50_) and 70% (IC_70_) were graphically determined.

### 2.5. Determination of Cell Death after Electrochemotherapy

One day before the experiment 5000 cells/well (B16-F10), 2500 cells/well (4T1), or 4000 cells/well (CT26) in 100 µL/well were seeded in µ-Slide 18 Well (Ibidi GmbH, Gräfelfing, Germany). Before electrochemotherapy, the cell culture medium was replaced with cell culture medium without phenol red (Gibco), and the cells were imaged using a Cytation 1 cell imaging multimode reader (BioTek, Agilent Technologies, Inc., Santa Clara, CA, USA). Then, the cell culture medium was changed to medium for electrochemotherapy containing the IC_30_, IC_50_ or IC_70_ concentration of chemotherapeutics followed by electroporation (two stainless steel plate electrodes with 2.4 mm distance between the electrodes, 8 square wave pulses, 1300 V/cm, 100 µs duration at frequency of 1 Hz). The cell culture medium was again changed 5 min after electrochemotherapy with fresh cell culture medium without phenol red (Gibco). The cells were then imaged at 0 h, 4 h, and 24 h after electrochemotherapy with Cytation 1 (BioTek). Cells were then counted using Gen 5 software (BioTek).

### 2.6. Determination of Extracellular ATP

For ATP determination, electrochemotherapy was performed as for the clonogenic assay. For the timepoints 0 and 4 h, the cells were incubated in 24 well ultra-low attachment plates (Corning Costar) and at the designated times 250 µL of cell culture medium was collected. For the 24 and 48 h timepoints, the cells were seeded on 6 well plates (Corning Costar) after electrochemotherapy and at the designated timepoints 250 µL of cell culture medium was collected. The samples were then centrifuged (300× *g*, 4 °C, 5 min) and 10 µL of the supernatant was used in the ATP Determination Kit (Invitrogen, Thermo Fisher Scientific) assay following the manufacturer’s instructions. The resulting luminescence was measured with Cytation 1 (BioTek), and the final ATP concentration was calculated from the obtained standard curve.

### 2.7. Determination of HMGB1 Release

For the determination of HMGB1 concentration, electrochemotherapy was performed as for the clonogenic assay. For the timepoint of 4 h, the cells were incubated in 24-well ultra-low attachment plates (Corning Costar) after electrochemotherapy and at the designated timepoint cells were collected in Protein LoBind tubes (Eppendorf, Hamburg, Germany). For the 24 and 48 h timepoints, the cells were seeded on T25 flasks (VWR International) after electrochemotherapy and at the designated timepoints, cells were collected in Protein LoBind tubes. The samples were then centrifuged (500× *g*, 4 °C, 5 min), and cell supernatants were collected in new Protein LoBind tubes and frozen at −80 °C until further analysis. Cell culture medium that was intended for collection 48 h after electrochemotherapy was changed with fresh cell culture medium after 24 h. At every collection time point, total volume of cell culture media was measured, and cells were counted. The HMGB1 concentration in undiluted cell culture medium was detected with an HMGB1 Detection Kit (Chondrex, Inc., Woodinville, WA, USA), following the manufacturer’s instructions. The final HMGB1 concentration was normalized to 1 × 10^6^ cells.

### 2.8. RNA Isolation and Quantitative PCR (qPCR)

For the determination of CD40 expression, 90% confluent cells growing in T75 flasks (VWR International) were harvested, and RNA was extracted using a peqGOLD Total RNA Kit (VWR International) following the manufacturer’s instructions. The RNA concentration was determined by spectrophotometry with Cytation 1 (BioTek) at 260 nm and its purity by measuring the absorbance ratios at 260/280 nm and 260/230 nm. Next, 2000 ng of total RNA was reverse transcribed to cDNA using the SuperScript VILO cDNA Synthesis Kit (Thermo Fisher Scientific) according to the manufacturer’s instructions. After reverse transcription, 10 ng of cDNA template was used in 20 µL of qPCR using predesigned primers: Ms.PT.58.41522911 (CD40) (Integrated DNA Technologies, Inc., Coralville, IA, USA) or custom-made primers: F-CTGTGCTGTCCCTGTATGC, R-GGCACAGTGTGGGTGAC (β-actin) (Integrated DNA Technologies) and PowerUp™ SYBR™ Green Master Mix (Thermo Fisher Scientific). The qPCRs were run until the 40th cycle with the following conditions: 2 min at 50 °C, 2 min at 95 °C, 40 cycles of 15 s at 95 °C, and 1 min at 60 °C; for the melting curve determination, 15 s at 95 °C, 1 min at 60 °C, and 1 min at 95 °C. The results were analyzed on a QuantStudio™ 3 Real-Time PCR System (Thermo Fisher Scientific). A Ct value above the 35th cycle was considered undetermined. Relative expression was calculated by normalization to the expression of the housekeeping gene β-actin using ΔCt method.

### 2.9. Determination of Cell Surface Markers by Flow Cytometry

For the determination of cell surface markers MHC I, MHC II, PD-L1, CRT, and CD40, electrochemotherapy was performed as for the clonogenic assay. For the timepoint of 4 h, the cells were incubated in 24-well ultra-low attachment plates (Corning Costar) and at the designated time, the cells were strained through 40 µm nylon strainers (Falcon, Corning), centrifuged and washed once with 1× PBS. For the 24 and 48 h timepoints, the cells were seeded on T75 flasks (VWR International) after electrochemotherapy, and at the designated timepoints the cells were dissociated with Versene Solution (Gibco), centrifuged, washed once with 1× PBS, and counted to obtain 1 × 10^6^ cells per staining.

The cells were then stained for MHC I, MHC II, PD-L1, CRT, and CD40 with the fixable viability dye e780 (Thermo Fisher Scientific) (LIVE/DEAD) for 30 min on ice in the dark (list of antibodies in Appendix A and gating strategy in Appendix A). The cells were then washed twice with Annexin V binding buffer and stained with FITC Annexin V (BioLegend, San Diego, CA, USA) for 30 min on ice in the dark. Afterwards, the cells were washed once in 1× PBS, fixed in IC Fixation buffer (Thermo Fisher Scientific) and analyzed with BD FACS Symphony A3 flow cytometer. Cells were first identified based on forward and side scatter, followed by exclusion of doublets. Then, the cells were gated for CRT translocation (e780−, CRT+ population) or apoptosis (e780−, Annexin V+ population), necrosis (e780+, Annexin V+ population) and live cells (e780−, Annexin V− population). From the live cell gate, the relative frequency (% of cells) and median fluorescence intensity (MFI) were determined for each of the interrogated markers. The MFI values were then normalized to the MFI values in the control (Ctrl) group. Data were analyzed using FlowJo software (Tree Star Inc., Ashland, OR, USA).

### 2.10. Statistics

All values in this study represent the mean ± SEM. All graphical presentations and statistical analyses were made in GraphPad Prism 9 (GraphPad Software). Comparison of means was performed with one-way ANOVA or nested ANOVA followed by Dunnett’s multiple comparisons test. Differences were considered significant at * *p* ≤ 0.05, ** *p* ≤ 0.01, *** *p* ≤ 0.001, **** *p* ≤ 0.0001. Sample size (n) represents biological replicates for each experiment and is presented in each figure legend.

## 3. Results

### 3.1. Cell Permeability and Survival

To ascertain that the results obtained in our study are not due to the differences in the electropermeabilization efficacy among the three cell lines used, we first evaluated the effect of electric pulses alone (8 pulses, 100 µs, 1 Hz) with different voltage-to-distance ratios on cell membrane permeability. All three tested cell lines were comparably permeable at all tested voltage-to-distance ratios (Figure 1A), reaching a plateau at 1300 V/cm. Furthermore, the SF of cells treated with 1300 V/cm voltage-to-distance ratio did not significantly differ among the three cell lines and was in the range of 70–80% compared to the control untreated cells (Figure 1B). As the 1300 V/cm is also the voltage-to-distance ratio used in the clinical electrochemotherapy setting [39,40,41]; thus, we used it in the rest of our study.

Because we were interested in the immunological effects of electrochemotherapy at three different drug concentrations (IC_30_, IC_50_, IC_70_) we first determined the cytotoxicity of electrochemotherapy with CDDP (Figure 1C–F), OXA (Figure 1G–J) and BLM (Figure 1K–N). From the obtained SF curves, the IC_30_, IC_50_, and IC_70_ values were determined (Appendix A). All three cell lines were equally sensitive to CDDP and BLM, as there was no statistically significant difference in IC_30_, IC_50_ and IC_70_ between them (Figure 1D–F, L–N). In contrast, the 4T1 and CT26 cell lines were more sensitive to OXA, as the IC_30_, IC_50_ and IC_70_ concentrations were significantly lower than those in the B16-F10 cell line (Figure 1H–J).

### 3.2. Cell Death after ECT

Next, the type of cell death after electrochemotherapy was determined with flow cytometry analysis of Annexin V, LIVE/DEAD stained cells at 4, 24 and 48 h after electrochemotherapy with CDDP, OXA and BLM at IC_30_, IC_50_ and IC_70_ concentrations. Interestingly, in all examined conditions and timepoints, majority of the cells were still viable (LIVE/DEAD−, Annexin V− population), with the percent of live cells ranging between 60 and 95% (Figure 2A–I). Nonetheless, the percentage of necrotic (LIVE/DEAD+, Annexin V+ population) and apoptotic (LIVE/DEAD−, Annexin V+ population) cells increased with higher concentrations used and later timepoints in all cell lines, independent of the chemotherapeutic used. The highest percentage of necrotic and apoptotic cells was determined in the B16-F10 cell line (Figure 2A–C, Appendix A), followed by the 4T1 cell line (Figure 2D–F, Appendix A), with the CT26 cell line having the lowest percentage of necrotic and apoptotic cells after electrochemotherapy (Figure 2G–I, Appendix A). There were no statistically significant differences in the mode of cell death among the three chemotherapeutics used in any of the cell lines (Figure 2D–I). Due to greater graph clarity the statistics are shown in Appendix A.

Because the observed differences in necrotic and apoptotic populations did not recapitulate the IC values determined with the clonogenic assay, a time-lapse imaging experiment with the IC_50_ concentrations was performed to delineate the effect of electrochemotherapy in the initial hours after the treatment. In all three cell lines, the percent of cells immediately after electrochemotherapy significantly decreased to ~20% of pre-electrochemotherapy values (Figure 3A–F and Appendix A). This was accompanied with swelling and rounding of some of the remaining attached cells, which is also the population of cells that was assayed with flow cytometry. Moreover, there was almost no additional change in the number of observed cells within 24 h after electrochemotherapy (Figure 3A–C), indicating that the proliferation of the electrochemotherapy treated cells was severely impeded and that the remaining cells did not die during the observation period.

### 3.3. Changes in DAMPs

#### 3.3.1. Electrochemotherapy Induced Release of ATP Decreases with Time

Among the three cell lines, the B16-F10 cell line released the lowest amount of ATP after electrochemotherapy compared to control cells regardless of chemotherapeutic and concentration used (Figure 4A–C and Appendix A). The 4T1 (Figure 4D–F and Appendix A) and CT26 (Figure 4G–I and Appendix A) cell lines released similar amounts of ATP after electrochemotherapy in all experimental conditions. In all three cell lines and with all three concentrations used, the highest release of ATP was determined within 4 h after electrochemotherapy, with the B16-F10 (Figure 4A and Appendix A) and CT26 (Figure 4G and Appendix A) reaching the maximum value at 0 h (immediately after electrochemotherapy) and the 4T1 cell line at 4 h after electrochemotherapy (Figure 4D and Appendix A). In B16-F10 cell line, a significantly decreased post-electrochemotherapy release of ATP was determined for all chemotherapeutics and concentrations used except for BLM IC_70_ (Appendix A). Similarly, in the 4T1 cell line, a statistically significant increase in release of ATP was determined for all chemotherapeutics and concentrations used except for CDDP IC_50_ (Appendix A), and in the CT26 cell line, a decrease was recorded for all chemotherapeutics and concentrations used (Appendix A). Due to greater graph clarity the statistics is shown in Appendix A.

#### 3.3.2. Electrochemotherapy Induced Release of HMGB1 Is Most Prominent at 24 h after Electrochemotherapy

Release of HMGB1 from cells differs from ATP release, and occurs at later time points. Therefore, we focused only on the 4, 24 and 48 h timepoints after electrochemotherapy. As expected, the highest changes in the released HMGB1 compared to control cells were determined 24 h after electrochemotherapy or later (Figure 5A–I). None of the cell lines exhibited an increase in release of HMGB1 at 4 h after treatment. In the B16-F10 cell line, the significantly increased release of HMGB1 was detected 24 h after electrochemotherapy with the IC_50_ concentration of CDDP (Figure 5B) and at 24 and 48 h after electrochemotherapy with the IC_50_ concentration of OXA (Figure 5B,C). In contrast, in the 4T1 cell line all three chemotherapeutics caused an increase in HMGB1 release 24 h after electrochemotherapy at IC_50_ concentrations (Figure 5E). Additionally, an increase in HMGB1 release was also determined for electrochemotherapy with the CDDP IC_70_ concentration at this timepoint (Figure 5E). Furthermore, this increased release of HMGB1 was still present at 48 h after electrochemotherapy with all three chemotherapeutics at the IC_30_ and IC_50_ concentrations, and for CDDP at IC_70_ concentration (Figure 5F). In contrast, in the CT26 cell line an increased release of HMGB1 was observed only at 48 h after electrochemotherapy with CDDP IC_70_ (Figure 5G–I). Thus, HMGB1 release after electrochemotherapy is most prominent at 24 h after treatment and with CDDP. Moreover, the 4T1 cell line had the highest increase in the release of HMGB1, followed by the B16-F10 cell line, whereas HMGB1 release was almost absent in the CT26 cell line.

#### 3.3.3. Electrochemotherapy Causes CRT Translocation to the Cell Surface

In the B16-F10 cell line, the translocation of CRT to the cell surface was first detected at 24 h after electrochemotherapy with all three chemotherapeutics at the IC_70_ concentration and after electrochemotherapy with OXA at IC_50_ (Figure 6A,B and Appendix A). The highest percentage of CRT+ cells was determined 48 h after electrochemotherapy with OXA and BLM at IC_50_ and IC_70_ concentrations (Figure 6C and Appendix A). Interestingly, electrochemotherapy with CDDP did not cause translocation of CRT at 48 h after the treatment (Figure 6C and Appendix A). In contrast, in the 4T1 cell line the translocation of CRT to the cell membrane was determined already at 4 h after electrochemotherapy with all three chemotherapeutics at all three tested concentrations (Figure 6D and Appendix A). At later timepoints, only electrochemotherapy with CDDP was able to induce the translocation of CRT at the IC_70_ concentration at 24 and 48 h (Figure 6E,F and Appendix A) after treatment and electrochemotherapy with OXA IC_70_ at 24 h (Figure 6E and Appendix A). In the CT26 cell line, the highest percent of CRT+ cells was determined at 24 h after electrochemotherapy with CDDP and OXA at the IC_50_ and IC_70_ concentration and at the IC_30_ concentration for CDDP (Figure 6G–I and Appendix A). The translocation of CRT to the cell membrane was also determined at 48 h after electrochemotherapy with CDDP at IC_30_ and IC_70_ concentrations and with OXA at the IC_70_ concentration (Figure 6I and Appendix A). Interestingly, electrochemotherapy with BLM did not result in the translocation of CRT in the CT26 cell line (Figure 6G–I and Appendix A).

### 3.4. Changes in Cell Surface Markers

#### Electrochemotherapy Increases MHC I Expression

In general, the expression of MHC I was increased in all three cell lines at 24 to 48 h after electrochemotherapy (Figure 7A–I and Appendix A), but not at 4 h (Figure 7A,D,G and Appendix A). The most prominent change was observed in the B16-F10 cell line where all three chemotherapeutics at IC_50_ and IC_70_ induced the increased expression of MHC I already at 24 h after electrochemotherapy. In the case of BLM, at this time point, BLM elicited the MCH I increase also at IC_30_ concentration of the drug (Figure 7B and Appendix A). At 48 h after electrochemotherapy, MHC I expression was increased by BLM at all three concentrations and OXA at IC_50_ and IC_70_ of (Figure 7C and Appendix A). In the 4T1 cell line, electrochemotherapy with all three chemotherapeutics increased the expression of MHC I at 24 h after electrochemotherapy but only at IC_70_ (Figure 7E and Appendix A). Interestingly, at 48 h after electrochemotherapy, the IC_30_ concentration of all tested chemotherapeutic increased the MHC I expression, but only CDDP was effective at IC_50_ and IC_70_ (Figure 7F and Appendix A). In the CT26 cell line, CDDP and OXA at IC_70_ increased the expression of MHC I at 24 h after electrochemotherapy (Figure 7H and Appendix A). At 48 h after electrochemotherapy, an increase in MHC I expression was determined for CDDP at all three concentrations used, for OXA at IC_50_ and IC_70_ and for BLM at IC_70_ (Figure 7I and Appendix A). Of note, a slight decrease in MHC I expression was determined in the 4T1 cell line at 4 h after electrochemotherapy with all three chemotherapeutics at the IC_50_ concentration (Figure 7D and Appendix A) and in the CT26 cell line at 24 h after electrochemotherapy for the BLM IC_30_ concentration (Figure 7H and Appendix A).

### 3.5. Electrochemotherapy Changes MHC II Expression

The expression of MHC II generally decreased in all three cell lines within 24 h after electrochemotherapy (Figure 8A–I and Appendix A). In the B16-F10 cell line, the decrease in MHC II expression was detected already at 4 h after electrochemotherapy with all three chemotherapeutics at IC_70_ and for CDDP at IC_50_ (Figure 8A), but not at later timepoints (Figure 8B,C). However, a higher percentage of MHC II expressing B16-F10 cells was observed with all three chemotherapeutics 4 h after electrochemotherapy with IC_50_ and IC_70_ concentrations and at 24 h after electrochemotherapy with IC_70_ concentrations (Appendix A).

Similarly, in the 4T1 cell line, the decrease in MHC II expression was noted at 4 h after electrochemotherapy with all three chemotherapeutics at IC_70_ and for OXA and BLM at IC_50_. However, unlike in the B16-F10 cell line, the decrease in MHC II expression at IC_70_ lasted up to 48 h after electrochemotherapy with all three chemotherapeutics used (Figure 8B,C and Appendix A). In line with this observation, there was no increase in the percentage of MHC II expressing cells after electrochemotherapy with any of the chemotherapeutics. At 24 h after electrochemotherapy with the IC_30_ concentration of BLM and with the IC_70_ concentrations of CDDP, OXA and BLM, a decrease in the percentage of MHC II expressing cells was observed (Appendix A). On the other hand, in the CT26 cell line, the decrease in MHC II expression was determined only at 24 h after electrochemotherapy with all three chemotherapeutics at IC_30_ and IC_70_ concentrations (Figure 8G–I and Appendix A). In line with this observation, there was no increase in the percentage of MHC II expressing cells after electrochemotherapy with any of the chemotherapeutics. At 24 h after electrochemotherapy with IC_30_ concentrations of CDDP, OXA and BLM even a decrease in the percentage of MHC II expressing cells was observed (Appendix A).

### 3.6. Electrochemotherapy Increases PD-L1 Expression

The expression of PD-L1 increased in all three cell lines within 24 h after electrochemotherapy (Figure 9A–I and Appendix A). In the B16-F10 cell line, the initial decrease at 4 h after electrochemotherapy in PD-L1 expression was detected with all three chemotherapeutics and at all concentrations used (Figure 9A). However, at 24 h after electrochemotherapy the PD-L1 expression was increased by all tested drugs at IC_30_ and IC_50_, and by OXA also at IC_70_ (Figure 9B and Appendix A). At 48 h, it was increased for OXA and BLM at all three concentrations used (Figure 9B,C and Appendix A). In the 4T1 cell line, an increase in PD-L1 expression was observed at 24 and 48 h after electrochemotherapy, but not at 4 h (Figure 9D–F and Appendix A). In the CT26 cell line, all three chemotherapeutics caused an increase in PD-L1 expression 24 h after electrochemotherapy at all three concentrations used (Figure 9H and Appendix A). However, at 48 h after electrochemotherapy, the same was observed at IC_50_ and IC_70_, but at IC_30_ only the PD-L1 expression was increased only by CDDP (Figure 9I and Appendix A).

### 3.7. Electrochemotherapy with CDDP and OXA Increases CD40 Expression

Because the B16-F10 and CT26 cell lines do not express CD40, as confirmed by qPCR, (Figure 10A) the experiments were performed only in the 4T1 cell line. electrochemotherapy with all three chemotherapeutics increased CD40 expression 24 h after electrochemotherapy at IC_70_ concentrations (Figure 10C and Appendix A), but only CDDP at this concentration was effective at 4 and 48 h as well (Figure 10B–D and Appendix A). Additionally, CDDP at IC_30_ induced increased expression of CD40 at 24 and 48 h after electrochemotherapy and OXA at IC_30_ and IC_50_ at 48 h after electrochemotherapy (Figure 10D and Appendix A).

## 4. Discussion

Electrochemotherapy is an effective local ablative antitumor therapy used in treatment of superficial and deep-seated tumors and is currently used in 170 centers around the world [42,43,44,45]. It enables targeted delivery of anticancer drugs to cells exposed to electric pulses, which makes them more effective at lower doses [2]. The capacity of electrochemotherapy to induce ICD mainly depends on the chemotherapeutic used but also on the treated tumor type [32,46]. During ICD, dying tumor cells release DAMPs that stimulate an adaptive immune response against tumor neo-antigens, thus turning them into an anticancer vaccine [46]. In this study, we evaluated the potential of electrochemotherapy with the clinically relevant chemotherapeutics CDDP, OXA and BLM to induce ICD-associated DAMPs and to cause changes in the expression of other immunologically important cellular markers in tumor cells in vitro. The pre-apoptotic translocation of CRT from the endoplasmic reticulum to the cell surface in dying cells is a key event of ICD [47]. CRT exposure is accompanied by ATP secretion and the post-apoptotic release of HMGB1 into the environment [14,19]. In our study, electrochemotherapy with all three tested chemotherapeutics induced ICD-associated DAMPs, but the induced DAMP signature was cell line and chemotherapeutic concentration specific. Some ICD-triggering regimens may also boost antigenicity through the modification of cell surface marker expression [25,34,37,38]. In our study, we showed that electrochemotherapy with CDDP, OXA or BLM can modify the expression of MHC I, MHC II, PD-L1 and CD40 cell surface markers. Similarly, as for DAMPs, the potential of electrochemotherapy to change their expression was cell line and chemotherapeutic concentration specific (Figure 11).

ICD can be induced by different stressors, including several chemotherapeutic drugs. OXA and BLM are well established ICD inducers, whereas the ICD inducing potential of CDDP is still contested [18,34]. The majority of the research with all three chemotherapeutics was done without electroporation as a delivery method, thus the concentrations used were much higher than those used in electrochemotherapy. Calvet et al. [10] showed that electrochemotherapy with a much lower concentration of BLM than that used in conventional systemic chemotherapy can induce ICD in vitro in CT26 cells [10]. Therefore, in our study, we first determined the IC_30_, IC_50_ and IC_70_ concentrations of all three chemotherapeutics for electrochemotherapy in three different cell lines. We have selected the three different cell lines to cover the two most commonly treated tumor types with electrochemotherapy, B16-F10 cells, as a model of human melanoma, 4T1 cells, as a model of human breast cancer, and the CT26 colon adenocarcinoma cell line as one of the most widely used cell lines in the studies of the anti-tumor immune response. We showed that all three cell lines were equally sensitive to CDDP and BLM, whereas B16-F10 cells were slightly more resistant to OXA than CT26 and 4T1 cells. The determined IC values cover the range of CDDP, OXA and BLM concentrations that are commonly used in electrochemotherapy in vitro [10,48,49,50]. With this we set the stage to directly compare the ICD inducing potential of electrochemotherapy with CDDP, OXA or BLM in the three tested cell lines. 

Using the determined IC values for individual chemoterapeutic, we then investigated the mode and timeline of cell death after electrochemotherapy. It was previously shown that after electrochemotherapy with CDDP, OXA, or BLM cells can die through apoptosis, necrosis, necroptosis, or pyroptosis, depending on the chemotherapeutic and cell line employed [51,52,53,54,55]. However, a detailed examination of the timeline of modes of cell death has not yet been performed. We thus used flow cytometry analysis of Annexin V (a marker of apoptosis) and live/dead marker stained cells at different times after electrochemotherapy to determine the timeline of apoptosis and necrosis after electrochemotherapy with CDDP, OXA, or BLM. We showed that the predominant mode of cell death in our setting was necrosis, although apoptosis was also present, which mainly started 24 to 48 h after electrochemotherapy, regardless of the chemotherapeutic or cell line used. Similarly, Calvet et al. [10] showed that the onset of cell death after electrochemotherapy with BLM in the CT26 cell line was 40 h after electrochemotherapy. This is consistent with the mechanism of action of CDDP, OXA and BLM where the majority of cells die due to induced DNA damage during their division [7,8,9]. However, as the percent of necrotic and apoptotic cells did not match the determined IC values, we examined the behavior of cells few minutes after electrochemotherapy. Using time-lapse microscopy of cells before and after electrochemotherapy, we determined that a substantial percentage of cells die immediately after electrochemotherapy and that the surviving cells remain in cell cycle arrest for at least 24 h after electrochemotherapy. This would indicate that there are two separate peaks of cell death after electrochemotherapy *in vitro.* The first one occurs immediately after electrochemotherapy and is probably connected to the increased uptake of the chemotherapeutic drug and membrane perturbations [56]. The onset of second one is between 24–48 h after electrochemotherapy, being connected to the cytotoxic action of chemotherapeutic drugs delivered into cells with EP [10,57]. Whether this happens also in vivo setting still needs to be investigated.

In line with this observation is the decreased ATP release, after the initial increase, determined after electrochemotherapy with CDDP, OXA, or BLM observed in the present study. We showed that the highest decrease in ATP release occurred either few minutes after electrochemotherapy, which was the case in the B16-F10 and CT26 cell lines, or after 4 h post-electrochemotherapy as recorded in the 4T1 cell line. The timeline of ATP release was not dependent on the used chemotherapeutic used or its concentration. However, the statistically significant changes in ATP release was drug-, timepoint- and cell line- specific (Figure 11A). The most potent inducer of ATP release was OXA at IC_70_ concentration, followed by CDDP where the IC_30_ concentration in the B16-F10 cell line and IC_50_ and IC_70_ concentrations in the 4T1 cell line failed to induce ATP release. The least potent proved to be electrochemotherapy with BLM with IC_70_ in B16-F10 cell line, and IC_50_ and IC_70_ in 4T1 cell line (Figure 11A). In our study, we only followed ATP release up to 48 h after electrochemotherapy, and within this timeframe, we did not observe another peak of ATP release that would coincide with the second wave of cell death. This observation differs from previous one where electrochemotherapy with BLM resulted in increase in ATP release 30 h after electrochemotherapy; however, a higher concentration (50 nM) of BLM was used in that study compared to ours (0.01–1.8 nM) [10].

Next, we investigated the release of HMGB1 from cells after electrochemotherapy. Generally, in the process of ICD, HMGB1 is released from cells later than ATP; therefore, we focused only on the 4, 24 and 48 h timepoints after electrochemotherapy. As expected, the highest changes in the released HMGB1 compared to control cells were determined at 24 h after electrochemotherapy or later (Figure 5). In this case, we determined a less uniform response compared to ATP (Figure 11A). The least responsive cell line was CT26, where HMGB1 release was observed only after electrochemotherapy with IC_70_ CDDP. This differs to published results where electrochemotherapy with BLM in the CT26 cell line; however, the concentrations used in our experiments were substantially lower [10]. B16-F10 cells were more responsive, and HMGB1 release was observed after electrochemotherapy with IC_50_ CDDP and IC_50_ OXA; we did not observe changes in the HMGB1 release after electrochemotherapy with BLM. In contrast, in the 4T1 cell line, only electrochemotherapy with IC_70_ OXA and IC_70_ BLM failed to induce HMGB1 release from the cells. Although it was already demonstrated for CDDP, OXA and BLM that they induce HMGB1 release, the concentrations used in our study are substantially lower than previously reported. Indeed, we used electroporation to enhance their entry but in non-electrochemotherapy setting a prolonged incubation with high concentrations of drugs might act differently on the HMGB1 release than in our study with short exposure to smaller concentrations employed.

CRT translocation from the ER to the cell surface is one of the most important hallmarks of ICD. Similarly, as with HMGB1, we focused only on time points 4, 24 and 48 h after electrochemotherapy. In our study, we demonstrated that electrochemotherapy with CDDP and OXA can induce translocation of CRT in all three cell lines, whereas electrochemotherapy with BLM fails to induce it in the CT26 cell line (Figure 11A). Similarly, in the study of Calvet et al. [10], electrochemotherapy with BLM failed to increase CRT translocation in the CT26 cell line at 0 h after electrochemotherapy. However, even after extending this time frame further, we did not prove that CRT translocate even at 48 h after electrochemotherapy. Additionally, we did show that electrochemotherapy with BLM can induce CRT translocation in B16-F10 and 4T1 cell lines, indicating that this response is cell line specific. We further provide evidence that electrochemotherapy with CDDP and OXA is also a potent inducer of CRT translocation in all three cell lines used, indicating a more ubiquitous usability of these two chemotherapeutics as a potential ICD inducers when used in the context of electrochemotherapy.

In addition to the emission of DAMPs, several ICD-triggering regimens may also boost antigenicity. Among the important cell surface receptors responsible for boosting antigenicity are MHC I, MHC II, PD-L1 and CD40 [25,27,28,29,30,31]. Reports of the effect of electrochemotherapy on these receptors are scarce. Ursic et al. [58] demonstrated that electrochemotherapy with OXA can increase the surface expression of MHC I in B16-F10 and 4T1 cell lines and electrochemotherapy with BLM in the B16-F10 cell line. Additionally, electrochemotherapy with BLM increased the surface expression of PD-L1 in the B16-F10 cell line. Of note, electrochemotherapy with CDDP had no effect on MHC I or PD-L1 surface expression in B16-F10, 4T1 or CT26 cell lines [58]. In our study, we performed a detailed examination of how electrochemotherapy with CDDP, OXA or BLM modifies four different cell surface receptors important for boosting antigenicity. We demonstrated that electrochemotherapy with all three chemotherapeutics used increased MHC I and PD-L1 expression in all three cell lines (Figure 11B). MHC I is important for presentation of antigens, including neo-antigens on the cell surface for recognition by immune cells [25]. Therefore, in our study we demonstrate that electrochemotherapy with CDDP, OXA or BLM, in addition to killing tumor cells, can also improve their recognition by the immune system, through the increased representation of tumor neoantigens by increased expression of surface MHC I receptors. In contrast, a high expression of PD-L1 on surface tumor cells is a poor prognostic marker in several tumor types, e.g., squamous cell carcinoma of the head and neck, melanoma, thyroid, thymus, esophagus, lung, breast, gastrointestinal tract, ovary, skin and others [59]; therefore, our results demonstrate that electrochemotherapy with CDDP, OXA and BLM is also a double-edged sword, as increased PD-L1 expression can counteract the positive effects of ICD on the activation of the immune system, by blocking the cytotoxic activity of CD8+ cytotoxic T cells [59,60,61]. In light of these results, a combination of immune checkpoint inhibitors (ICIs) in concert with electrochemotherapy would be a prudent approach. Indeed, several reports on electrochemotherapy with BLM and ICI confirmed an improved tumor response to combined therapy [11,58,62].

Regarding MHC II expression after electrochemotherapy, it seems to be more cell line specific than chemotherapeutic drug- specific. The only cell line in which we determined increased MHC II expression after electrochemotherapy with CDDP, OXA and BLM was the B16-F10 cell line, though not at all concentrations of the drugs used (Figure 11B). On the other hand, we even determined a decrease in the expression of MHC II in the 4T1 and CT26 cell lines after electrochemotherapy with CDDP, OXA or BLM. The main role of MHC II receptors is to present processed antigens, which are derived primarily from exogenous sources, to CD4+ T-lymphocytes. Therefore, they are critical for the initiation of the antigen-specific immune response [26]. In light of this fact, knowledge of the MHC II specific response of the treated cell line or tumor to electrochemotherapy is important when activation of the antitumor immune response is anticipated or interpreted. In the study of Ursic et al., immunotherapy with gene electrotransfer of plasmid DNA encoding IL-12 in combination with electrochemotherapy with CDDP, OXA or BLM was more effective in B16-F10 tumors than in 4T1 and CT26 tumors [58]. The authors attributed the alterations in response to differences in the initial immunogenicity of the tumors. However, considering our results on MHC II expression in vitro, a potential better mounting of the antitumor immune response in the B16-F10 cell line, compared to the 4T1 and CT26 tumors, might be a key determinant.

Finally, we demonstrated that CD40 is expressed only in the 4T1 cell line and not in the B16-F10 or CT26 cell line. In the 4T1 cell line the expression of CD40 increased after electrochemotherapy with all chemotherapeutics, except at IC_30_ and IC_50_ BLM. The CD40 is important costimulatory cell surface receptor and mediator of the immune response [26,30,31]. Therefore, the higher expression of CD40 after electrochemotherapy in the 4T1 cell line is a favorable event speaking in favor of the immune-modulating effects of electrochemotherapy with CDDP, OXA or BLM.

In our study, we examined the behavior of several DAMPs and cell surface receptors after electrochemotherapy with CDDP, OXA or BLM at different timepoints, but only until 48 h after electrochemotherapy: This is one of the limitations of our study, as we have determined that the second wave of cell death starts at the end of our observation period. Therefore, it is possible that important changes in the observed markers happen later than 48 h after electrochemotherapy. Moreover, although we have showed that ICD is induced in vitro after electrochemotherapy with CDDP, OXA or BLM, this does not answer the question whether it also happens in vivo. Our results should therefore be further explored in an in vivo setting, to confirm that ICD is indeed induced after electrochemotherapy with CDDP, OXA or BLM and at which concentrations of chemotherapeutics. This is especially important when chemotherapeutic is delivered systemically, such as the case of BLM, because due to the non-functional tumor vasculature, the local concentration of the chemotherapeutic can vary between different tumor regions. Further, because we only performed in vitro experiments, we cannot conclude whether ICD induced by electrochemotherapy with CDDP, OXA or BLM would be effective in mounting an anti-tumor immune response. Our results also show that the ICD-inducing potential of chemotherapy with CDDP, OXA or BLM is cell line dependent; therefore, experiments using relevant human cell lines should be performed to validate the translational potential of our results. Nevertheless, this study brings new important insights into the induction of ICD by electrochemotherapy with CDDP, OXA or BLM that set the corner stone for future in vivo validation of the observed effects in mouse tumor models in immunocompetent mice and later translation into clinical settings.

## 5. Conclusions

In this study, we evaluated the potential of electrochemotherapy with the clinically relevant chemotherapeutics CDDP, OXA and BLM to induce ICD-associated DAMPs and to cause changes in the expression of other immunologically important cellular markers in tumor cells in vitro. We showed that electrochemotherapy with all three tested chemotherapeutics induced ICD-associated DAMPs, but the induced DAMP signature was cell line and chemotherapeutic concentration specific. Moreover, we showed that electrochemotherapy with CDDP, OXA or BLM can modify the expression of MHC I, MHC II, PD-L1 and CD40 cell surface markers that are important in boosting the immunogenicity of the therapy. Similarly, as for DAMPs, the potential of electrochemotherapy to change their expression was cell line and chemotherapeutic concentration specific. Our results thus put the electrochemotherapy with clinically relevant chemotherapeutics CDDP, OXA and BLM on the map of ICD inducing therapies and provide the necessary mechanistic insights needed to harness the potential of ICD to elicit a systemic anti-tumor immune response, especially when combined with immunotherapies.

## Figures and Tables

**Figure 1 vaccines-11-00925-f001:**
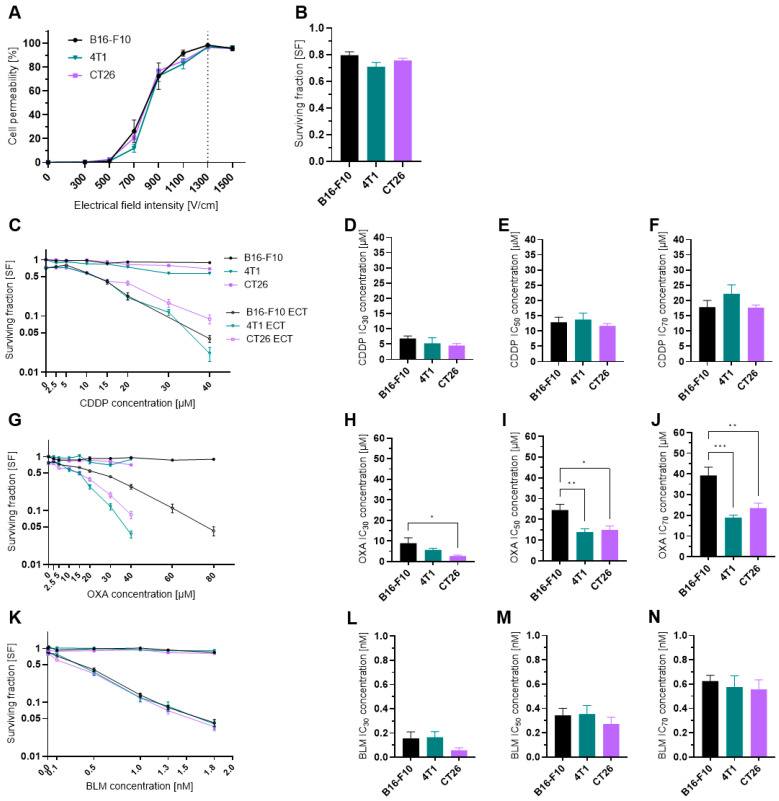
Cell permeability and cytotoxicity after ECT. (**A**) Cell permeability at different electrical field intensities. *n* = 3–4. (**B**) Surviving fraction of cells treated only with electrical pulses at 1300 V/cm. *n* = 18. (**C**) Surviving fraction of cells treated with CDDP or ECT with CDDP. *n* = 6–13. (**D**) Determined CDDP IC_30_ concentrations in µM. *n* = 5–6. (**E**) Determined CDDP IC_50_ concentrations in µM. *n* = 5–6. (**F**) Determined CDDP IC_70_ concentrations in µM. *n* = 5–6. (**G**) Surviving fraction of cells treated with OXA or ECT with OXA. *n* = 5–9. (**H**) Determined OXA IC_30_ concentrations in µM. n = 5–6. (**I**) Determined OXA IC_50_ concentrations in µM. *n* = 5–6. (**J**) Determined OXA IC_70_ concentrations in µM. *n* = 5–6. (**K**) Surviving fraction of cells treated with BLM or ECT with BLM. *n* = 5–6. (**L**) Determined BLM IC_30_ concentrations in µM. *n* = 4–6. (**M**) Determined BLM IC_50_ concentrations in µM. *n* = 5–6. (**N**) Determined BLM IC_70_ concentrations in µM. *n* = 5–6. The values are presented as the AM ± SEM. * *p* ≤ 0.05, ** *p* ≤ 0.01, *** *p* ≤ 0.001. Where not indicated with *, differences are not significant.

**Figure 2 vaccines-11-00925-f002:**
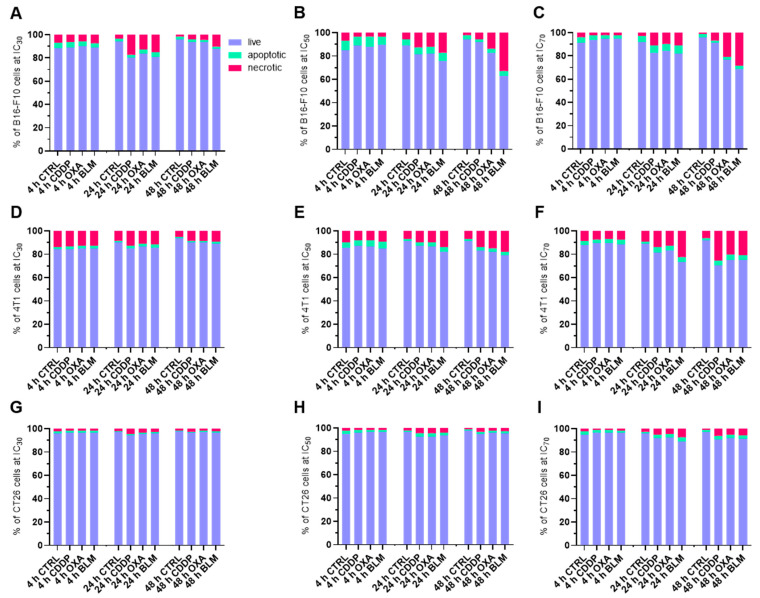
Percent of live (LIVE/DEAD−, Annexin V− population), apoptotic (LIVE/DEAD−, Annexin V+ population) and necrotic (LIVE/DEAD+, Annexin V+ population) cells after ECT. Cell populations present in B16-F10 cell line samples after ECT with CDDP, OXA or BLM at 4, 24 and 48 h after ECT at (**A**) IC_30_, (**B**) IC_50_, and (**C**) IC_70_. Cell populations present in 4T1 cell line samples after ECT with CDDP, OXA or BLM at 4, 24 and 48 h after ECT at (**D**) IC_30_, (**E**) IC_50_, and (**F**) IC_70_. Cell populations present in CT26 cell line samples after ECT with CDDP, OXA or BLM at 4, 24 and 48 h after ECT at (**G**) IC_30_, (**H**) IC_50_, and (**I**) IC_70_. The values are presented as the AM; *n* = 4.

**Figure 3 vaccines-11-00925-f003:**
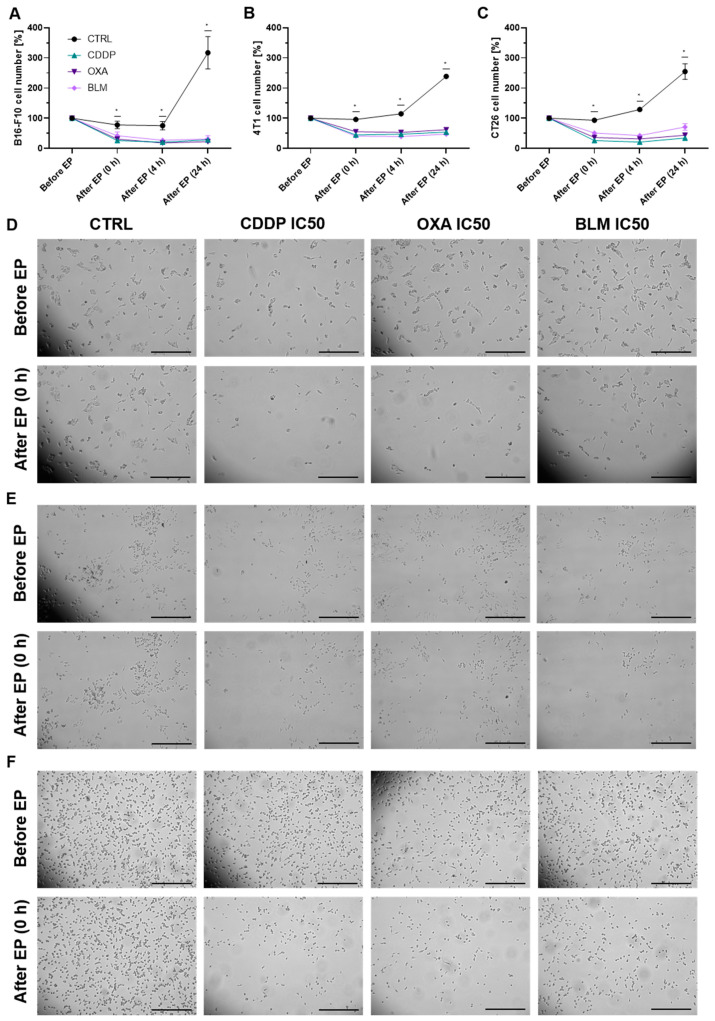
Changes in cell number after ECT. (**A**) Cell number in the B16-F10 cell line before and after ECT with IC_50_ concentrations of CDDP, OXA or BLM. (**B**) Cell number in the 4T1 cell line before and after ECT with IC_50_ concentrations of CDDP, OXA or BLM. (**C**) Cell number in the CT26 cell line before and after ECT with IC_50_ concentrations of CDDP, OXA or BLM. The values are presented as the AM ± SEM. * *p* ≤ 0.05 compared to control, *n* = 3. (**D**) Representative images of B16-F10 cells before and immediately after ECT with IC_50_ concentrations of CDDP, OXA or BLM. (**E**) Representative images of 4T1 cells before and immediately after ECT with IC_50_ concentrations of CDDP, OXA or BLM. (**F**) Representative images of CT26 cells before and immediately after ECT with IC_50_ concentrations of CDDP, OXA or BLM. Scale bar: 0.5 mm.

**Figure 4 vaccines-11-00925-f004:**
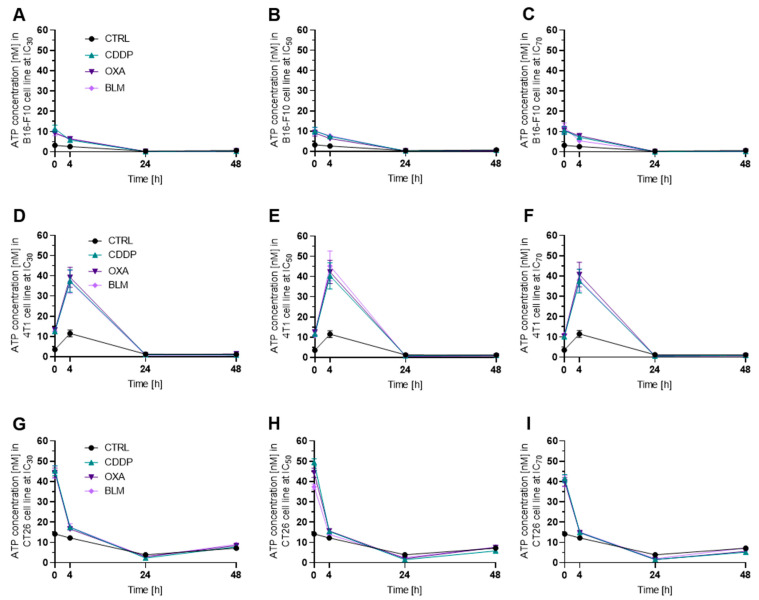
ATP release after ECT. ATP release in the B16-F10 cell line after ECT with (**A**) IC_30_, (**B**) IC_50_ and (**C**) IC_70_ concentrations of CDDP, OXA or BLM. ATP release in the 4T1 cell line after ECT with (**D**) IC_30_, (**E**) IC_50_ and (**F**) IC_70_ concentrations of CDDP, OXA or BLM. ATP release in the CT26 cell line after ECT with (**G**) IC_30_, (**H**) IC_50_ and (**I**) IC_70_ concentrations of CDDP, OXA or BLM. The values are presented as the AM ± SEM; *n* = 3.

**Figure 5 vaccines-11-00925-f005:**
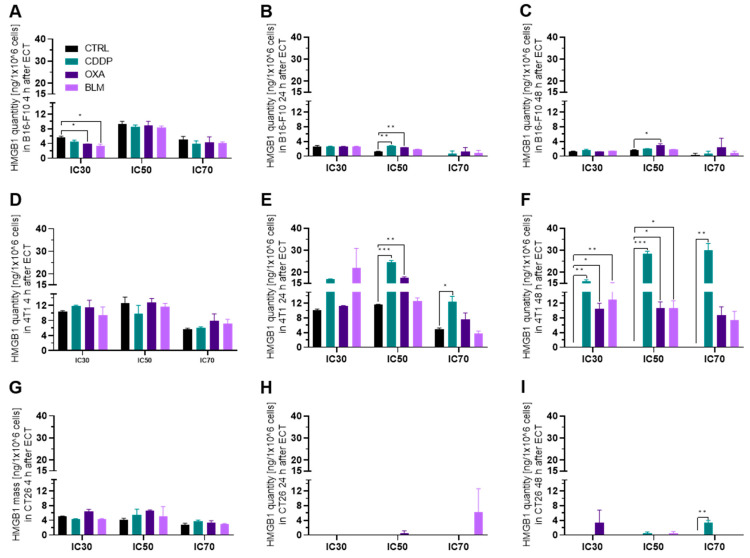
HMGB1 release after ECT. HMGB1 release in the B16-F10 cell line (**A**) 4 h, (**B**) 24 h and (**C**) 48 h after ECT with IC_30_, IC_50_ or IC_70_ concentrations of CDDP, OXA or BLM. HMGB1 release in the 4T1 cell line (**D**) 4 h, (**E**) 24 h and (**F**) 48 h after ECT with IC_30_, IC_50_ or IC_70_ concentrations of CDDP, OXA or BLM. HMGB1 release in the CT26 cell line (**G**) 4 h, (**H**) 24 h and (**I**) 48 h after ECT with IC_30_, IC_50_ or IC_70_ concentrations of CDDP, OXA or BLM. The values are presented as the AM ± SEM. * *p* ≤ 0.05, ** *p* ≤ 0.01, *** *p* ≤ 0.001, *n* = 2.

**Figure 6 vaccines-11-00925-f006:**
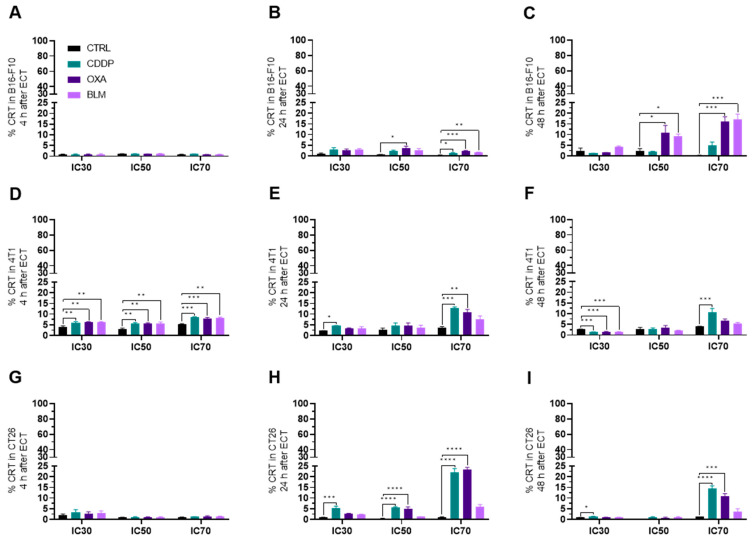
CRT translocation after ECT. Percentage of CRT+ cells in the B16-F10 cell line (**A**) 4 h, (**B**) 24 h and (**C**) 48 h after ECT with CDDP, OXA or BLM. Percentage of CRT+ cells in the 4T1 cell line (**D**) 4 h, (**E**) 24 h and (**F**) 48 h after ECT with CDDP, OXA or BLM. Percentage of CRT+ cells in the CT26 cell line (**G**) 4 h, (**H**) 24 h and (**I**) 48 h after ECT with CDDP, OXA or BLM. The values are presented as the AM ± SEM. * *p* ≤ 0.05, ** *p* ≤ 0.01, *** *p* ≤ 0.001, **** *p* ≤ 0.0001, *n* = 4.

**Figure 7 vaccines-11-00925-f007:**
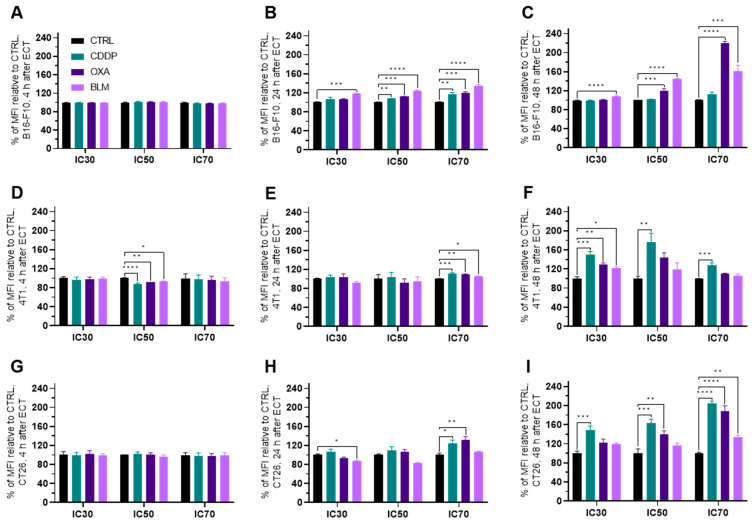
MHC I expression after ECT presented as a percent of MFI relative to Ctrl. MHC I expression in the B16-F10 cell line (**A**) 4 h, (**B**) 24 h and (**C**) 48 h after ECT with CDDP, OXA or BLM. MHC I expression in the 4T1 cell line (**D**) 4 h, (**E**) 24 h and (**F**) 48 h after ECT with CDDP, OXA or BLM. MHC I expression in the CT26 cell line (**G**) 4 h, (**H**) 24 h and (**I**) 48 h after ECT with CDDP, OXA or BLM. The values are presented as the AM ± SEM. * *p* ≤ 0.05, ** *p* ≤ 0.01, *** *p* ≤ 0.001, **** *p* ≤ 0.0001, *n* = 4.

**Figure 8 vaccines-11-00925-f008:**
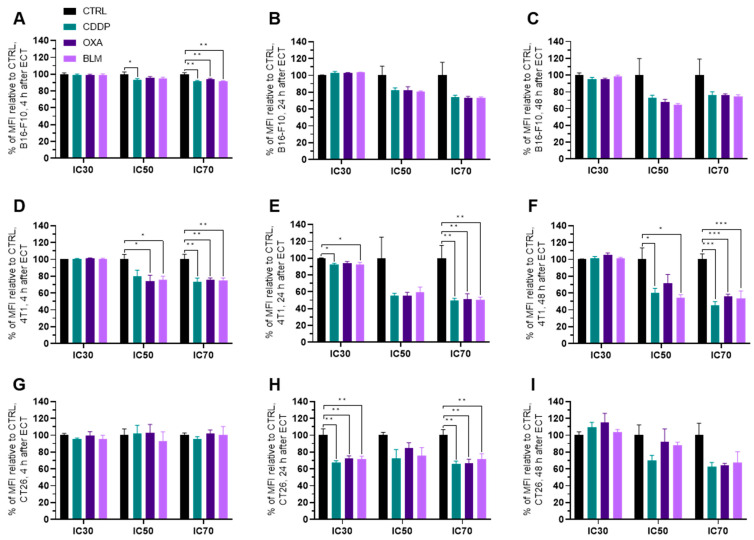
MHC II expression after ECT presented as a percent of MFI relative to Ctrl. MHC II expression in the B16-F10 cell line (**A**) 4 h, (**B**) 24 h and (**C**) 48 h after ECT with CDDP, OXA or BLM. MHC II expression in the 4T1 cell line (**D**) 4 h, (**E**) 24 h and (**F**) 48 h after ECT with CDDP, OXA or BLM. MHC II expression in the CT26 cell line (**G**) 4 h, (**H**) 24 h and (**I**) 48 h after ECT with CDDP, OXA or BLM. The values are presented as the AM ± SEM. * *p* ≤ 0.05, ** *p* ≤ 0.01, *** *p* ≤ 0.001, *n* = 4.

**Figure 9 vaccines-11-00925-f009:**
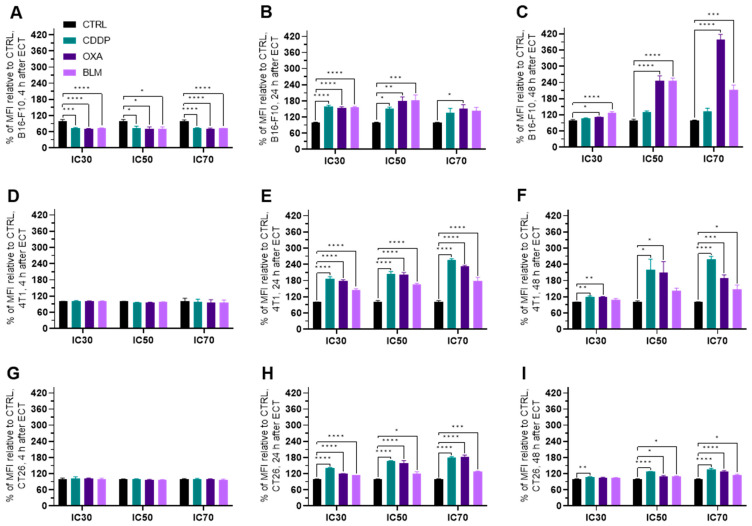
PD-L1 expression after ECT as a percent of MFI relative to Ctrl. PD-L1 expression in the B16-F10 cell line (**A**) 4 h, (**B**) 24 h and (**C**) 48 h after ECT with CDDP, OXA or BLM. PD-L1 expression in the 4T1 cell line (**D**) 4 h, (**E**) 24 h and (**F**) 48 h after ECT with CDDP, OXA or BLM. PD-L1 expression in the CT26 cell line (**G**) 4 h, (**H**) 24 h and (**I**) 48 h after ECT with CDDP, OXA or BLM. The values are presented as the AM ± SEM. * *p* ≤ 0.05, ** *p* ≤ 0.01, *** *p* ≤ 0.001, **** *p* ≤ 0.0001, *n* = 4.

**Figure 10 vaccines-11-00925-f010:**
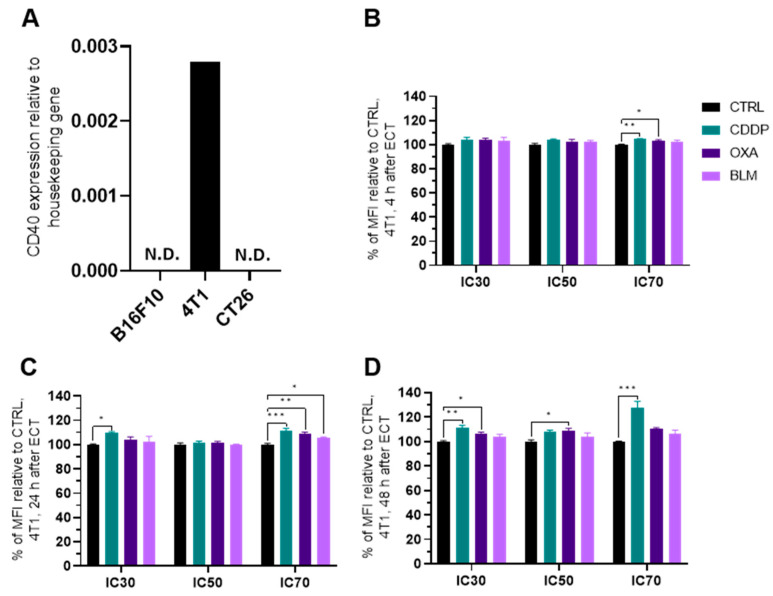
CD40 expression after ECT. (**A**) Expression of CD40 in different cell lines determined with qPCR. Percent of MFI increase relative to Ctrl in CD40 expression in the 4T1 cell line (**B**) 4 h, (**C**) 24 h and (**D**) 48 h after ECT with CDDP, OXA or BLM. The values are presented as the AM ± SEM. * *p* ≤ 0.05, ** *p* ≤ 0.01, *** *p* ≤ 0.001, *n* = 4. Where not indicated with *, differences are not significant. N.D.: not determined.

**Figure 11 vaccines-11-00925-f011:**
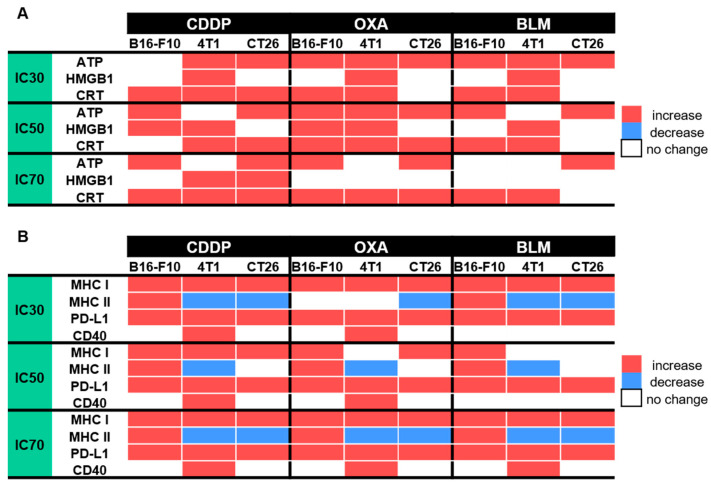
Summarized changes in the expression of specific DAMPs and cell surface markers within 48 h after ECT; (**A**) Changes in specific DAMPs after ECT with CDDP, OXA or BLM, (**B**) Changes in the expression of cell surface markers after ECT with CDDP, OXA or BLM.

## Data Availability

The data presented in this study are available on request from the corresponding author.

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
