# Peer review of "Effects of Electrochemotherapy on Immunologically Important Modifications in Tumor Cells"

_vaccines, 2023, doi:10.3390/vaccines11050925_

Round 1
Reviewer 1 Report
The study of Kesar et al. is a highly interesting, very well designed and comprehensive study showing important immunologically changes in tumor cells in response to electrochemotherapy. All results on DAMPs and surface markers are graphical summarized in an extra figure 11, which is greatly appreciated.
Author Response
Reviewer 1: The study of Kesar et al. is a highly interesting, very well designed and comprehensive study showing important immunologically changes in tumor cells in response to electrochemotherapy. All results on DAMPs and surface markers are graphical summarized in an extra figure 11, which is greatly appreciated.
We thank the reviewer for his positive comments and opinion about our manuscript.
Reviewer 2 Report
This research analyzed the effect of electrochemotherapy associated with anticancer drugs in 3 types of murine cell lines. The evaluated effect was on DAMPs, but also immune checkpoint markers such as MHC and PD-L1. These immune checkpoint markers, such as the PD-L1, are recently a hotspot in research because of the use of inhibitors in human diseases, including inflammatory conditions and cancer.
The manuscritpt is well written, it is easy to read, and to understand. There are many abbreviations, my recommendation would be to use less to help unfamiliar readers of this specific topic.
Electrochemotherapy has been used in skin conditions such as epidermolysis bullosa, but also in non-melanoma skin cancer such as classic Kaposi sarcoma.
There are many data in this report, I wonder why if only IC50 would suffice.
Additional comments:
(1) Could you please explain why murine cell lines were selected? Why not using human cell lines (for translation medicine)?
(2) Is it correct to mix in the experimental design melanoma, mammary carcinoma, and colorectal carcinoma? Were this subtypes selected because are relevant human cancers?
(3) In Figure 1 D to N. Could you please remind the variables that is being evaluated in the Y axis? Is it the survival fraction? Why the survival fraction is higher in IC70 than IC30?
(4) Line 314. What is the difference between apoptosis and necrosis? What markers were used to identify these two different mechanisms? This information could be added in Figure 2.
(5) In Figure 3. Apart form the number of cells, was the morphology also changed?
(6) Could you please explain why it is important to show the IC30 and IC70? Half-maximal inhibitory concentration (IC50) is the most widely used and informative measure of a drug's efficacy.
(7) In Figure 7D, IC50 is significant, but not in IC70. Could you please explain why?
(8) Is MFI the mean or median fluorescence intensity?
Author Response
Reviewer 2:
This research analyzed the effect of electrochemotherapy associated with anticancer drugs in 3 types of murine cell lines. The evaluated effect was on DAMPs, but also immune checkpoint markers such as MHC and PD-L1. These immune checkpoint markers, such as the PD-L1, are recently a hotspot in research because of the use of inhibitors in human diseases, including inflammatory conditions and cancer.
We thank the reviewer for his comments and suggestions. We have taken them into account and revised our manuscript accordingly
The manuscript is well written, it is easy to read, and to understand. There are many abbreviations, my recommendation would be to use less to help unfamiliar readers of this specific topic.
We have reconsidered our use of the abbreviations in our manuscript and have replaced ECT with electrochemotherapy throughout the manuscript. The rest of the abbreviations used in the manuscript are the common abbreviations in the field of immunogenic cell death (ICD) that were also used in the latest review articles from Galluzzi et al. (1) and Kroemer et al. (2) Therefore, we would like to keep them in order to use the same nomenclature as it is currently used in the field of ICD.
(1) Galluzzi L, Kepp O, Hett E, Kroemer G, Marincola FM. Immunogenic cell death in cancer: concept and therapeutic implications. J Transl Med [Internet]. 2023 Dec 1 [cited 2023 Apr 25];21(1):1–8. Available from: https://translational-medicine.biomedcentral.com/articles/10.1186/s12967-023-04017-6
(2) Kroemer G, Galassi C, Zitvogel L, Galluzzi L. Immunogenic cell stress and death. Nat Immunol 2022 234 [Internet]. 2022 Feb 10 [cited 2023 Apr 25];23(4):487–500. Available from: https://www.nature.com/articles/s41590-022-01132-2
Electrochemotherapy has been used in skin conditions such as epidermolysis bullosa, but also in non-melanoma skin cancer such as classic Kaposi sarcoma.
We thank the reviewer for this observation. We have now expanded the description of electrochemotherapy and its usage to state the following.
“Electrochemotherapy (ECT) is a local ablative technique, mainly employed in treatment of superficial tumors, with published clinical guidelines supporting its use in melanoma, squamous cell carcinoma, including its management in patients with epidermolysis bullosa, breast cancer, Merkel cell carcinoma, soft tissue sarcomas and bone metastases.(3)”
And added a new reference that covers the latest development in the use of electrochemotherapy in superficial tumors.
(3) Campana LG, Miklavčič D, Bertino G, Marconato R, Valpione S, Imarisio I, et al. Electrochemotherapy of superficial tumors – Current status:: Basic principles, operating procedures, shared indications, and emerging applications. Semin Oncol. 2019 Apr 1;46(2):173–91.
There are many data in this report, I wonder why if only IC50 would suffice.
We thank the reviewer for this observation and we agree that often the IC50 would suffice. However, as we also show in our manuscript, the induction of different DAMPs and cell surface markers is dependent on the concentration of the chemotherapeutic used. Most importantly, at higher concentrations the immunogenic cell death (ICD) inducing potential of the chemotherapeutics could be lost, due to prevailing mechanism of death being necrosis rather than ICD. Therefore, we have decided to show the effects of the IC30, IC50 and IC70 to show, that the ICD-inducing potential of electrochemotherapy with all three tested chemotherapeutics is indeed concentration dependent.
Additional comments:
(1) Could you please explain why murine cell lines were selected? Why not using human cell lines (for translation medicine)?
We agree with the reviewer that human cell lines have a better translational potential than murine cell lines in majority of cases. However, as one of the key mechanisms of the immunogenic cell death is its capacity to activate and induce the anti-tumor immune response, it is unfortunately still necessary to use murine cell lines that can grow in immunocompetent mice. Although substantial progress has been made with humanized mouse models, the use of murine cell lines in combination with immunocompetent mice is still predominant experimental approach when anti-tumor immune response is being studied. Therefore, we believe that our use of murine cell lines is appropriate to support the execution of the next translational step, in vivo studies in mouse tumor models in immunocompetent mice, where our results provide the key information necessary for proper planning of the studies. To acknowledge this limitation of our study, we have modified the last paragraph of the discussion to now state the following:
“… Our results also show that the ICD-inducing potential of chemotherapy with CDDP, OXA or BLM is cell line dependent; therefore, experiments using relevant human cell lines should be performed to validate the translational potential of our results. Nevertheless, this study brings new important insights into the induction of ICD by electrochemotherapy with CDDP, OXA or BLM that set the corner stone for future in vivo validation of the observed effects in mouse tumor models in immunocompetent mice and later translation into clinical settings.”
(2) Is it correct to mix in the experimental design melanoma, mammary carcinoma, and colorectal carcinoma? Were this subtypes selected because are relevant human cancers?
We thank the reviewer for this observation. Indeed, we have chosen two (B16-F10 and 4T1) out of the three cell lines as they represent the most commonly treated human tumors with electrochemotherapy. We have selected the CT26 cell line, as it is one of the most often used cell lines in the studies of anti-tumor immune response. With the selection of these cell lines we were aiming to provide a direct comparison within the manuscript of the ICD-inducing potential of chemotherapy among the most commonly treated tumor types and the “gold-standard” cell line that is used in these types of studies. We have also added this information to the manuscript in the discussion section:
“… We have selected the three different cell lines to cover the two most commonly treated tumor types with electrochemotherapy, B16-F10 cells, as a model of human melanoma, 4T1 cells, as a model of human breast cancer, and the CT26 colon adenocarcinoma cell line as one of the most widely used cell lines in the studies of the anti-tumor immune response. …”
(3) In Figure 1 D to N. Could you please remind the variables that is being evaluated in the Y axis? Is it the survival fraction? Why the survival fraction is higher in IC70 than IC30?
The units are stated in square brackets. Therefore, the concentrations to reach IC70 values are higher than to reach IC30 values.
In Figure 1 the panels A, B, C, G and K have the surviving fraction (SF) on the y axis. The rest of the panels (D, E, F, H, I, J, L, M and M) have the concentration of the drug at different IC values on the Y axis. We have provided the unit [µM] on each y axis as well. To further clarify this, we have added the information about the units in the figure legend as well. Considering this, the IC70 values are indeed higher than IC30 values, which is what it would be expected.
(4) Line 314. What is the difference between apoptosis and necrosis? What markers were used to identify these two different mechanisms? This information could be added in Figure 2.
We detected apoptosis and necrosis using flow cytometry analysis of LIVE/DEAD (Fixable Viability Dye eFluor™ 780) and Annexin V cells. Annexin V is a standard marker for labelling apoptotic cells when used in combination with LIVE/DEAD stain. We have gated our populations in the following way: live cells (LIVE/DEAD-, Annexin V- population), apoptotic cells (LIVE/DEAD-, Annexin V+ population) and necrotic cells (LIVE/DEAD+, Annexin V+ population). We have now also added the information about the different populations into the text, which now states the following:
… “Interestingly, in all examined conditions and timepoints, majority of the cells were still viable (LIVE/DEAD-, Annexin V- population), with the percent of live cells ranging between 60 and 95% (Figure 2A-I). Nonetheless, the percentage of necrotic (LIVE/DEAD+, Annexin V+ population) and apoptotic (LIVE/DEAD-, Annexin V+ population) cells increased …”
And also, in the figure legend, which now states the following:
“Percent of live (LIVE/DEAD-, Annexin V- population), apoptotic (LIVE/DEAD-, Annexin V+ population) and necrotic (LIVE/DEAD+, Annexin V+ population) cells after ECT. …”
(5) In Figure 3. Apart from the number of cells, was the morphology also changed?
We thank the reviewer for the comment. Indeed, we have noticed some cell swelling and rounding; however, we were not able to quantified this, as the resolution and quality of the images was not sufficient to properly quantify the observed changes. For this reason, we have also avoided describing it in the manuscript, as we believed that we didn’t have enough of high-quality data to definitively describe the changes. We have now modified the paragraph describing this to state the following:
“… This was accompanied with swelling and rounding of some of the remaining attached cells, which is also the population of cells that was assayed with flow cytometry. …”
(6) Could you please explain why it is important to show the IC30 and IC70? Half-maximal inhibitory concentration (IC50) is the most widely used and informative measure of a drug's efficacy.
We agree with the reviewer that in majority of cases the half-maximal inhibitory concentration (IC50) is the most widely used and informative measure. However, as we have stated above, we believe that we show in our manuscript, that the induction of different DAMPs and cell surface markers is dependent on the concentration of the chemotherapeutic used. Most importantly, at higher concentrations the immunogenic cell death (ICD) inducing potential of the chemotherapeutics could be lost, due to prevailing mechanism of death being necrosis rather than ICD. Therefore, we have decided to show the effects of the IC30, IC50 and IC70 to show, that the ICD-inducing potential of electrochemotherapy with all three tested chemotherapeutics is indeed concentration dependent.
(7) In Figure 7D, IC50 is significant, but not in IC70. Could you please explain why?
We thank the reviewer for this observation. Indeed, we have observed a trend of slightly greater changes in the expression of cell surface markers in the 4T1 cell line at IC50 compared to IC70 at 4h after electrochemotherapy (Figures 7D, 8D, 9D). Currently, we do not know the reason for this; however, we speculate that the higher concentrations of the chemotherapeutics might initially cause more damage to the DNA, which would then need longer to repair, therefore their effect on the cell surface markers would be delayed. However, this would have to be experimentally confirmed, which was outside of the scope of our manuscript.
(8) Is MFI the mean or median fluorescence intensity?
The abbreviation MFI stands for median fluorescence intensity. This is specified in the Materials and Methods section in the manuscript, lines 267-269, chapter 2.9 (Determination of cell surface markers by flow cytometry):
“From the live cell gate, the relative frequency (% of cells) and median fluorescence intensity (MFI) were determined for each of the interrogated markers. The MFI values were then normalized to the MFI values in the control (Ctrl) group.”